# The Correlation between Volatile Compounds Emitted from *Sitophilus granarius* (L.) and Its Electrophysiological and Behavioral Responses

**DOI:** 10.3390/insects13050478

**Published:** 2022-05-20

**Authors:** Lijun Cai, Sarina Macfadyen, Baozhen Hua, Wei Xu, Yonglin Ren

**Affiliations:** 1State Key Laboratory of Ecological Pest Control for Fujian/Taiwan Crops, Institute of Applied Ecology, Fujian Agriculture and Forestry University, Fuzhou 350002, China; cai-lijun@live.cn; 2Key Laboratory of Plant Protection Resources and Pest Management, Ministry of Education, Northwest A&F University, Yangling, Xianyang 712100, China; huabzh@nwafu.edu.cn; 3Agriculture and Food, Commonwealth Scientific and Industrial Research Organisation, Acton, Canberra, ACT 2601, Australia; sarina.macfadyen@csiro.au; 4College of Science, Health, Engineering and Education, Murdoch University, Murdoch, WA 6150, Australia

**Keywords:** *Sitophilus granarius*, granary weevil, volatile organic compounds (VOCs), electroantennogram, behavioral assay

## Abstract

**Simple Summary:**

Postharvest loss has become a crucial issue for the grain supply chain. Storage of grain without losses in quality is a critically important aspect of global food security. The monitoring and detection of insect infestations in stored grain is essential to inform pest management decisions. Insect olfaction is a principal sensory modality for sensing semiochemicals from their external environment and regulates their behaviors. Some semiochemicals function as attractants or repellents, which could be used for insect surveillance and pest control. In this study, the granary weevil *Sitophilus granarius* (L.), was used as an example to evaluate volatile compounds released from the weevils’ and their initiated electrophysiological and behavioral responses. An improved understanding of *S.*
*granari**us* chemical ecology will lead to developing more efficient and environmentally friendly pest control strategies and technologies.

**Abstract:**

The granary weevil *Sitophilus granarius* (L.) is one of the most serious primary insect pests of stored products. When *S. granarius* present in grains, various volatile organic compounds are released as chemical signals which can be used to detect the insects. In this study, volatile chemical compounds released from *S. granarius* were analyzed using the headspace solid phase micro-extraction (HS-SPME) coupled with gas chromatography (GC)–mass spectrometry (MS) techniques. Two key compounds, 3-hydroxy-2-butanone and 1-pentadecene, were identified from mixed gender of *S. granarius* adults at high density. Moreover, both male and female adults showed dose-dependent electroantennography (EAG) responses to 3-hydroxy-2-butanone. In behavioral assays, *S. granarius* was attracted to 3-hydroxy-2-butanone at 0.001 µg/10 µL but repelled at 10 µg/10 µL or higher. *S. granarius* was consistently repelled by 1-pentadecene at concentrations at 0.001 and 1000 µg/10 µL. 3-hydroxy-2-butanone and 1-pentadecene have considerable potential to offer in the development of new approaches for the monitoring and management of this destructive stored grain insect pest.

## 1. Introduction

The granary weevil *Sitophilus granariu**s* (L.) (Coleoptera: Curculionidae) is one of the most serious primary stored grain insect pests [1]. As internal feeders, both adults and larvae of a *S. granarius* attack a variety of stored grain, including wheat, maize, barley, oats and rye. Its infestation not only results in direct weight loss and contamination, but also makes stored grains susceptible to secondary storage pests and toxicogenic fungi [1]. Infestation by weevils and fungi can significantly decrease the germination and viability of grains, and eventually decreases the nutritional and economic value of the grains. It can even pose health risks to people susceptible to allergies [2]. Furthermore, as an internal feeder, infestation by immature stages of *S. granarius* in bulk grains is very difficult to find until serious damage has occurred. 

Large scale, repetitive fumigation of stored grain insect pests with synthetic chemicals, such as methyl bromide and phosphine, is frequently used to control storage pests [3]. However, efficacy of fumigants is reduced with the development of pest resistance [4]. Environmental and human health concerns have increased the demand for safer methods of food protection. Restricted legislation limiting the use of fumigants and broad-spectrum contact insecticides have made the control of storage pests increasingly challenging in the global market [5]. Therefore, more effective and sustainable alternative control strategies are urgently needed.

Insects rely on chemical signals from the environment to locate food, hibernation sites, aggregate conspecific individuals, potential mates and oviposition, and to avoid dangerous or unsuitable habitats and hosts [6]. Therefore, targeted use of selected chemical information, so-called semiochemicals, has provided a promising way to implement the combined control and monitoring of pests [7,8]. Attempts have been made both to isolate and identify pheromones from pest insects and chemical volatiles from storage products to generate feasible attractants or repellents. One such material is an aggregation pheromone called sitophilate (*R**, *S**-1-ethylpropyl-2-methyl-3-hydroxypentanoate) produced by male *S**. granarius*, attract both sexes [9,10], and has long been commercially used in traps with either its homologues or food baits in the field [11]. In addition, both genders of *S. granarius* respond to various extracts and volatiles emitted by wheat [12], maize [13], carob pods and peanuts [14]. However, more work is needed due to pheromones and traps are affected by many environmental factors in postharvest stored grain [15]. 

Since attractants as ‘pulling’ semiochemicals employed alone are not sufficiently effective to offer a robust pest control, a preventative ‘pushing’ strategy using repellents is considered as an alternative method. The repellents can repel insects and provide an additional direct pest control approach, through deterring host and/or oviposition site selection by pests [16]. Propionic acid and several short-chain aliphatic ketones or aldehydes have been reported as repellents to *S. granarius* [16]. They can markedly reduce the orientation ability of granary weevils towards odors of wheat grains [14,16,17]. For *Tribolium castaneum* (Herbst), crowding and presence of dead insects can cause their repellency [18]. When inhabited resources are saturated, further arrivals of the western pine beetle, *Dendroctonus brevico**mis* LeConte, are deterred [19]. A similar phenomenon was observed in the granary weevil’s attempt to disperse when the population increased and became crowded in our lab culture. However, little attention has been paid to using *S. granarius* generated volatile chemicals rather than sitophilate for management purposes. 

The headspace analysis techniques have been widely used in food and flavor industries for the quality assessment of food products. They also play a role in determining ongoing spoilage of foods and characterizing volatiles specific to stored pest products [20]. The headspace solid phase micro-extraction (HS-SPME) coupled with gas chromatography (GC)–mass spectrometry (MS) techniques are a useful and convenient method to extract and examine pheromones and other volatile secretions of coleopteran insects [21,22,23]. A big advantage of HS-SPME technique is that volatile extraction can be carried out with test samples under relatively natural conditions, and without interference from confounding factors such as organic solvents and heat treatments [21].

This study aimed to identify the volatile organic compounds to be released from the stored grain insect pest, *S. granarius*, and further investigate the weevils’ responses to these compounds using the HS-SPME technique, coupled with a gas chromatography–mass spectrometer (GC–MS), electroantennography (EAG) techniques and behavioral bioassays. The result from this project will improve our understanding of *S. granarius* chemical ecology and may help us develop highly efficient and environmentally friendly pest control approaches for managing this stored grain insect pest.

## 2. Materials and Methods

### 2.1. Insect Materials

Tested *S.*
*granarius* culture was derived from CSIRO Entomology Culture Collection, Canberra, Australia [24]. Cultures were set up with adult *S. granarius* (2.8 g) reared on 800 g of organic whole wheat grain in 2 L glass culture jars, under dark conditions at 30 ± 1 °C and 65 ± 3% relative humidity. The insect cultures were incubated, until adults of the next generation emerged (4–5 weeks).

### 2.2. Reagents and Apparatus

The chemicals used as standards were purchased from Sigma-Aldrich Pty. Ltd., USA, and included methanol, ethanol, n-hexane, 2, 3-butanedione, 1-pentadecene and 3-hydroxy-2-butanone with ≥95% purity. One liter Erlenmeyer flasks (Bibby Sterilin, Staffordshire, UK, Cat. No. FE 1 L/3) were used for preparation of standards. The measured volume of each Erlenmeyer flask and inlet system was calculated from the weight of water (20 °C) required to fill the container and was used for calculations. Erlenmeyer flasks of 250 mL (Alltech Cat. No. 9535) were used for preparation of samples. Each flask was fitted with a Mininert valve, equipped with a Teflon-coated septum lid (Alltech Cat. No. 95326) gas sampling system. 

### 2.3. Collection of Volatiles 

For collection of the volatile chemicals released from *S. granarius* adults, the experiment conducted in an empty flask (250 mL) as control and treatments with *S. granarius* adults included mixed-gender adult 1, 5, 10, 20, 50 and 100 individuals used per flask, which were set up and maintained at 25 ± 0.5 °C and 55–60% RH. The test insects were kept on wet filter paper for 30 min, allowing them to crawl and clean the insect body, and the insects were then cleaned further by transferring them to dry filter paper. Thirty minutes were allowed for the insects to settle before the flasks were sealed. The sample flasks were sealed tightly, to prevent volatiles from escaping. Five replicates for each treatment and control were prepared. 

Preliminary studies showed that *S. granarius* adults at the highest population density (100 adults per 250 mL flask) survived without mortality, for at least five days. At the end of a five-day exposure, oxygen levels had reduced to about 16% with CO_2_ at 4%.

A sample collection of volatiles was carried out manually with an 85 μm CAR/PDMS fiber attached with a holder to the flask closure. It was selected as its high extraction efficiency for relatively small molecular weight compounds, according to the guide of fiber selection provided by the manufacturer. The fiber was inserted into the headspace of the flask and exposed for 3 h to the upper portion of the headspace of *S. granarius* just below the septum, without agitation. The headspace volatiles were sampled every 4 h over a 48 h time series, after the first hour of incubation. This helped both standardize sampling time and monitor possible changes in volatile compound profiles in sealed flasks within 48 h. There were five replicates for each treatment and a control. Periodic blank flasks were tested as procedural controls to confirm the stability of the system. At the end of the defined extraction time, the fiber was withdrawn from the headspace into a sampling needle. The fiber holder was then removed from the extraction flask and injected into a GC-FID inlet, allowing absorbed volatiles to desorb at 250 °C for 5 min under split-less mode. Desorbed volatiles were analyzed by GC and GC–MS.

### 2.4. GC and GC–MS Analysis

For investigation of the volatile compounds extracted with SPME, a Varian 3400 CX GC (Varian Instruments, Sunnyvale, CA, USA) coupled with a split-splitless injector, a ZB-WAXplus column (30 m × 0.32 mm i.d. × 0.25 μm film thickness) and a Flame Ionization Detector (FID) was utilized as described before [25]. The GC oven temperature program was the same as previously reported [25]: 35 °C for 8 min, increasing to 120 °C at a rate of 5 °C/min and held at 120 °C for 10 min; 250 °C was set for FID temperature and nitrogen was used as the carrier gas with a constant flow of 1.1 mL/min.

A Shimazu gas chromatograph mass spectrometer (GC–MS-QP2010 Plus) was used to analyze and identify that above detected volatile compounds. A Stabilwax^®^ Restek column (30 m × 0.25 mm i.d. × 0.25 μm film thickness) was used with helium as the carrier gas at a constant speed of 30 cm/s. The oven temperature program was the same as above but included one more increase from 120 to 245 °C at 15 °C/min and was held for 10 min. Samples were quantitively analyzed by scanning the mass range between 40 and 300 amu. The National Institute of Standards and Technology (NIST, Gaithersburg, MD, USA) spectral library database was utilized to identify compound peaks by mass spectra comparisons with standards and the retention times of known authentic standards (Sigma-Aldrich, St. Louis, MO, USA).

### 2.5. Olfactometer

The open arena olfactometer was a modified version of a walking bioassay used for observing weevil behavior as described [24]. The olfactometer was 0.4 m long and 0.2 m wide, comprising a main basement plate, a top cover plate with a common and two branch tunnels in a Y-shape. Each branch tunnel led to a pitfall trap, about 8 cm away from the diversion point of the common tunnel. The tunnels were all 0.8 mm in width for insects to move forward and/or backward. A source of compressed air provided a laminar airflow of 12 mL/min over the arena and an air exhaust evacuated contaminated air to the outside of the building. The test odorant stimuli were released by air flow from treatments sealed in 1.1 L flasks, which were connected to the branch tunnel of the olfactometer. The laboratory temperature and relative humidity were controlled at 25 ± 0.5 °C and 60 ± 5%, respectively.

### 2.6. Pitfall Bioassay

The weevils were left to move on the bare basement plate because of their excellent crawling ability on a smooth surface. At each trial, 10 *S. granarius* adults were put in a vial. The vial was placed in the center of the triangle start arena and held for 30 s for insects to acclimate. The weevils were released and given 5 min to respond. Insects were given a choice between a specific dose of the test stimulus and control, the latter of which was either methanol or n-hexane. At least 10 replicates were prepared and tested. Four concentrations of 3-hydroxy-2-butanone and 1-pentadecene were made through diluting the former with methanol and the latter with n-hexane to 0.001, 0.1, 10 and 1000 µg/10 µL. The number of individuals falling into the pitfall, as well as those making no choice, was counted. 

### 2.7. Electrophysiology

Electroantennography (EAG) (Syntech, Preetz, Germany) responses were examined from the whole *S. granarius* immobilized in a pipette tip, with head and antennae exposed under laboratory conditions as described above. Silver wires were used as electrodes. The tip of the reference electrode was inserted into the base of the antenna. The tip of the recording electrode was connected to the basal half of the distal antennal segment, the distal half of which was cut off to allow conduction between electrode and hemolymph. The DC potential was recorded on a computer using a custom-built amplifier, and an IDAC A/D converter and software (Autospike SCII; Syntech, Preetz, Germany). Filtered and humidified air was continuously passed over the mounted antenna at approximately 0.5 m/s. A Pasteur pipette containing a 0.5 cm^2^ piece of filter paper impregnated with a 10 μL solution of the test chemical in a defined amount was used to deliver the stimulus in a 2.5 mL puff during 0.2 s by means of a Syntech stimulus controller (CS-55 model, Syntech, Preetz, Germany). The outlet of the delivery pipette was set at a rectangle angle to the antennal axis at a distance of approximately 10 mm, ensuring the whole antenna was within the airstream. Nine male and nine female weevils around four weeks after emergence were tested in each experiment. Each weevil was presented with 10 μL of 0.001, 0.1, 10 and 1000 μg/L 3-hydroxy-2-butanone in randomized order to give dose–response curves. A control of hexane, the solvent, was tested at the beginning and at the end of each recording, and the EAG response was subtracted from the antennal responses to the above dilution series. The preparation, manipulators, probe, microscope and light source were all housed in a Faraday cage.

### 2.8. Monitoring of Carbon Dioxide and Oxygen

Carbon dioxide (CO_2_) and oxygen (O_2_) levels were monitored with Witt OXYBABY^®^ 6.0 (WIT-Gasetechnik GmbH & Co KG T, Witten, Germany). Accuracy < 0.1% O_2_ and <0.01% CO_2_. For evaluation of the effect on respiration during exposure, 100 adults were placed into the flask without food. The concentrations of carbon dioxide and oxygen were measured at timed intervals during five days of exposure. All insects were transferred to 100 mL bottles containing 20 g cultural wheat and continually observed for 5 days. Five replicates were used for this test.

### 2.9. Statistical Analyses

Statistical analyses of the behavioral responses to chemicals were performed with the statistical software SPSS. For each bioassay trial, a response index (RI) was calculated using RI = [(T − C)/Tot] × 100%, where T is the number responding to the treatment, C is the number responding to the control, and Tot is the total number of insects released [14]. When positive values of RI indicate attraction to the treatment, negative ones indicate being repelled. Analyses of variance (ANOVA) were carried out for re-observations for both the electrophysiological and behavioral experiments. The significance of the mean RI in each treatment of the two-choice pitfall bioassay was evaluated by the Student’s *t*-test for paired comparisons. The most significant positive or negative mean values of RI were first analyzed by an analysis of variance and subsequently ranked by the least significant difference (LSD) multiple range test (*p* = 0.05).

Two factors were tested initially (antennal response and dose of tested chemical) for the electrophysiological trials. Differences between means were tested for significance with Dunnett’s (comparison with a control) or Newman and Keuls’ (comparison by groups of means) tests.

## 3. Results

### 3.1. Volatile Profile of S. granarius

The volatile compounds from live *S. granarius* adults were collected using the HS-SPME technique and identified by GC–MS. The mass spectrum of the released volatile compounds was compared with the National Institute of Standards and Technology (NIST) Library. The accuracy was more than 98% for all four major identifiable peaks: ethanol, 2,3-butanedione, 3-hydroxy-2-butanone, and 1-pentadecene (Figure 1). Though a few other peaks were also detected from GC–MS analysis, their matching degrees to the NIST database were either very low (<98%), or their amounts were much lower than the four identified peaks. Therefore, only ethanol, 2,3-butanedione, 3-hydroxy-2-butanone and 1-pentadecene, were further investigated in this study.

During the 48 h continuous monitoring processes, no 3-hydroxy-2-butanone could be detected in the 250 mL flasks containing less than 50 individual weevils. The presence of 3-hydroxy-2-butanone started to occur from 42 h in trials containing 50 and 100 individuals per flask. However, when the number of tested weevils achieved 50, no consistent result could be obtained. Two of the five trials with 50 weevils failed to show the presence of 3-hydroxy-2-butanone at any time. In the other three trials with 100 *S. granarius*, all obtained consistent readings of peak areas for 3-hydroxy-2-butanone (Figure 2). Ethanol and 2,3-butanedione were identified as volatile compounds in trace amounts in each trial and at any time point. They probably resulted from residues of wheat culture medium present on the weevils, which have been detected before [25].

### 3.2. Behavioral Bioassays

Behavioral bioassays of granary weevil responses to 3-hydroxy-2-butanone and 1-pentadecene were carried out to study whether these two compounds could initiate the behavioral responses (Figure 3). The results showed that adult granary weevils responded positively to low concentrations of 3-hydroxy-2-butanone at 0.001 µg in 10 µL solvent, but it turned negative as the concentration was increased to 0.1 µg/10 µL. When the concentration of 3-hydroxy-2-butanone increased to 1000 µg/10 µL, the mean number of weevils that made a choice fell to two individuals. Granary weevils constantly responded negatively to 1-pentadecene, with the concentrations increasing from 0.001 to 1000 µg/10 µL. When the concentration reached 10 µg/10 µL (Figure 3), the number of weevils that made a choice dropped. The difference between responses for each of the two consecutive concentrations of 3-hydroxy-2-butanone was significant (*p* < 0.05).

### 3.3. Electroantennography (EAG) Responses of S. granarius to Volatile Stimuli

EAG was used for evaluating whether 3-hydroxy-2-butanone and 1-pentadecene prepared at various concentrations in hexane could cause the weevils’ electrophysiological responses. Both male and female *S. granarius* showed EAG responses to the four concentrations of 3-hydroxy-2-butanone tested (Figure 4), while no response was detected to 1-pentadecene. Male granary weevils showed the strongest response, 5.35 ± 0.82 mV, at 1000 µg/10 µL, and the response intensities decreased with concentrations in a declining trend, 3.66 ± 0.61 mV at 10 µg/10 µL, 1.92 ± 0.38 mV at 0.1 µg/10 µL, 0.95 ± 0.11 mV at 0.001 µg/10 µL (Figure 4). For female weevils, the highest intensity was recorded as 4.99 ± 0.64 mV, at 1000 µg/10 µL, 3.36 ± 0.48 mV at 10 µg/10 µL, 1.88 ± 0.46 mV at 0.1 µg/10 µL, 0.94 ± 0.26 mV at 0.001 µg/10 µL (Figure 4). The standard errors indicated that the differences of antennal responses to different concentrations were significant. However, there were no significant differences on EAG responses between male and female *S.*
*granarius* at each tested concentration of 3-hydroxy-2-butanone.

### 3.4. Monitoring Carbon Dioxide and Oxygen

The high densities of weevils can increase CO_2_ and decrease O_2_ in the flask, which may induce the production of 3-hydroxy-2-butanone and 1-pentadecene. The variation of CO_2_ and O_2_ concentrations in the 250 mL sealed flasks were measured and results are shown in Figure 5. For five days of exposure, the measured concentrations of CO_2_ increased and O_2_ decreased, in comparison with the CO_2_ and O_2_ concentration of 0.03% and 21% in ambient air. However, CO_2_ concentrations were increased by 2.5 ± 0.5% within the first 32 h exposure and then slowly increased. Oxygen concentrations were decreased by 2.5 ± 0.6% within the first 32 h exposure and then slowly increased.

## 4. Discussion

The headspace volatiles from live *S. granarius* adults in sealed flasks at different densities (1, 5, 10, 20, 50 and 100 individuals per flask) were collected using an HS-SPME technique and analyzed by GC–MS in order to investigate chemicals released by the adult weevils, which may be the chemical signals involved in the chemical communication. Several main volatile components in the headspace of *S. granarius* adults were identified with the potential bio-functions of two main components, 3-hydroxy-2-butanone and 1-pentadecene, on the behavior of this species and were explored with EAG and behavior bioassay techniques. 

The compound 3-hydroxy-2-butanone explored from this study has only recently been reported in relation to *S. granarius* [25] but no functional analysis was performed. A study carried out on the lobster cockroach, *Nauphoeta cinerea* (Olivier), suggested 3-hydroxy-2-butanone may be a male sex pheromone produced from the genital segment. In *N. cinerea*, headspace volatiles were collected with activated charcoal and whole gland secretion extracted using solvents was analyzed with GC–MS [22]. This confirmed the presence of 3-hydroxy-2-butanone.

Dense populations of flour beetles would normally produce substantial amounts of benzoquinones potentially to deter further arrival of con-species [26]. Our study showed that 3-hydroxy-2-butanone could only be detected when the population was at a high-density level (at least 50 individuals in 250 mL flask) and experiencing starvation for almost two days. Therefore, we concluded that it was likely an anti-aggregation pheromone or stress compound among *S. granarius* adults. Furthermore, in this study, 3-hydroxy-2-butanone could elicit behavioral and antennal responses in both sexes of *S. granarius* adults, which exhibited the greatest electrophysiological responses to the highest concentration. They were slightly attracted to the lowest concentration but intensively repelled by a higher one, indicating the proposed bio-function to be a repellent at high dose.

During five days of exposure, no dead insects were found. After five days of exposure, all insects were transferred to 100 mL bottles containing 20 g cultural wheat and were continually observed for five more days. In this instance, two dead insects were found. This species could survive without food for 6–8 days. Thus, lack of food would not cause death directly within the time length of the trials. 

Bioactivity of some short-chain aliphatic ketones and aldehydes has been explored, such as 2-pentanone, 2-hexanone, 2-heptanone, 2,3-butanedione and 2-hexenal, against *S. granarius* [16]. Compared with the pheromone sitophilate, the use of 3-hydroxy-2-butanone as a tool to monitor or control populations in stored grain ecosystems would certainly prove advantageous. It would be considered safe, widely accepted and used as a flavoring substance in foodstuffs. 

Ethanol and 2,3-butanedione has been reported in cereal grains [25]. These two compounds detected here were in low amounts, which varied insignificantly among different treatment flasks. Consequently, we assumed that they might be from residual traces of culture media adhering to the insects. A trace of the compound 1-pentadecene was also detected here, which had been suggested as a defensive secretion in *T. castaneum* [27]. We also found it was an indication of infestation by *S. granarius* (unpublished data). Although EAG investigations demonstrated that it did not stimulate an electrophysiological response, it might play a role in repelling granary weevils according to the behavior bioassay results. A recent study on *T. castaneum*, showed that the low concentration (0.0025%) of 1-pentadecene exhibited an attractive effect [28]. Concentrations from 0.005% to 0.013% showed a neutral effect, while concentrations above 0.02% repelled *T. castaneum* [28]. This may be the same case for *S. granarius*. Another possibility is that 1-pentadecene may be detected by the *S. granarius* gustatory system (e.g., tarsi, ovipositor) but not the olfactory system (antennae). Therefore, 1-pentadecene did not initiate strong EAG responses. For example, *Aedes aegypti* mosquitoes have been reported using their legs to sense repellent DEET on contact [29].

Sitophilate, the commonly known aggregation pheromone of granary weevils, was not detected here. Three reasons might explain its absence. On the one hand, the screening and selection characteristics of the SPME fiber adopted for extraction might lead to the absence of sitophilate. On the other hand, aggregation pheromones would function more as an arresting chemical rather than an attractant chemical [30], suggesting that its secretion might not take place when food resources are unavailable. In this study, no food resource was provided. Lastly, sitophilate is known to be very unstable and to degrade quickly [30].

In summary, this study demonstrates that granary weevils release volatile chemicals likely acting as anti-aggregation pheromones. The HS-SPME technique is able to detect some characteristic volatile chemicals from storage insects under relatively natural conditions. The finding of 3-hydroxy-2-butanone is of significance from a management perspective. Further studies are necessary focus on evaluation of the repelling effect of 3-hydroxy-2-butanone on *S. granarius*. The behavior impact of 3-hydroxy-2-butanone on other storage product insects utilizing the same resources can be further investigated to highlight the possibility for this compound to also be a repellent rather than an attractant. How the *S. granarius* olfactory system detects 3-hydroxy-2-butanone at molecular level is also an interesting field for further study. For example, understanding which odorant receptor or odorant binding protein plays a critical role in sensing 3-hydroxy-2-butanone in *S. granarius* will shed insight into its olfactory system and provide a new molecular target for pest control [31].

## Figures and Tables

**Figure 1 insects-13-00478-f001:**
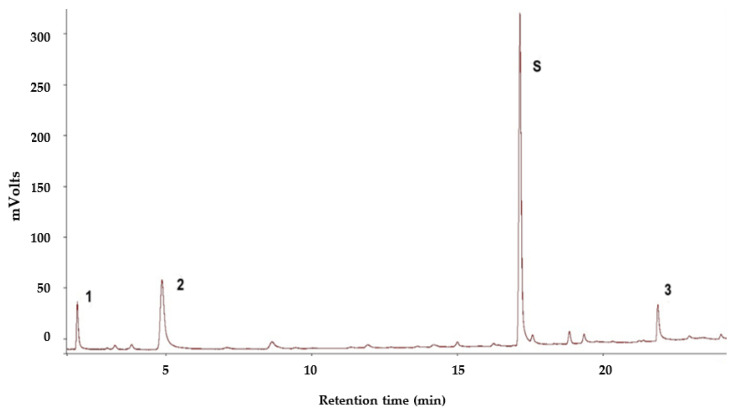
GC chromatograph of headspace volatiles from *S. granarius*. Numbered peaks: 1 = ethanol; 2 = 2,3-butanedione; S = 3-hydroxy-2-butanone; 3 = 1-pentadecene.

**Figure 2 insects-13-00478-f002:**
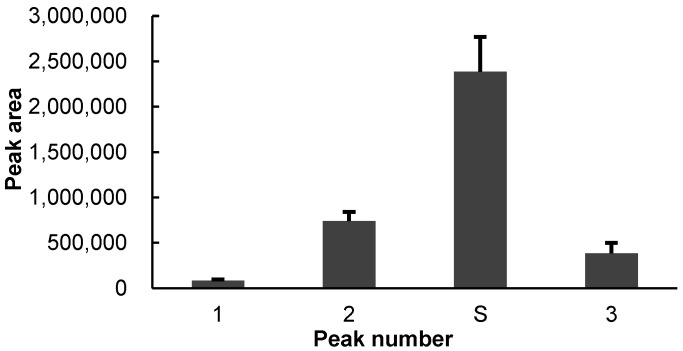
The GC peak areas of main peaks from *S. granarius* collected at 46–48 h after being sealed in an airtight container. Numbered peaks: 1 = ethanol; 2 = 2,3-butanedione; S = 3-hydroxy-2-butanone; 3 = 1-pentadecene. Each bar represents the average of 5 replicates and the error bars indicate standard deviation.

**Figure 3 insects-13-00478-f003:**
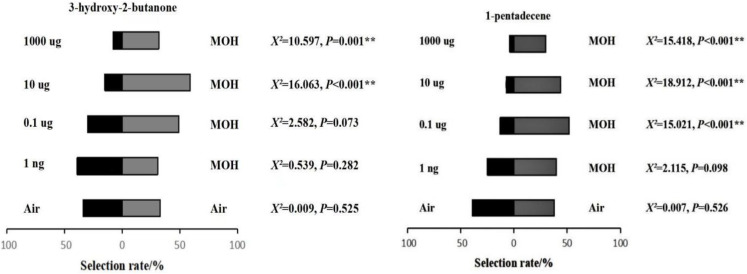
Mean behavioral responses (±) of *S. granarius* (*n* = 10) to 3-hydroxy-2-butanone and 1-pentadecene. ** indicate significant difference at the 0.01 level, respectively.

**Figure 4 insects-13-00478-f004:**
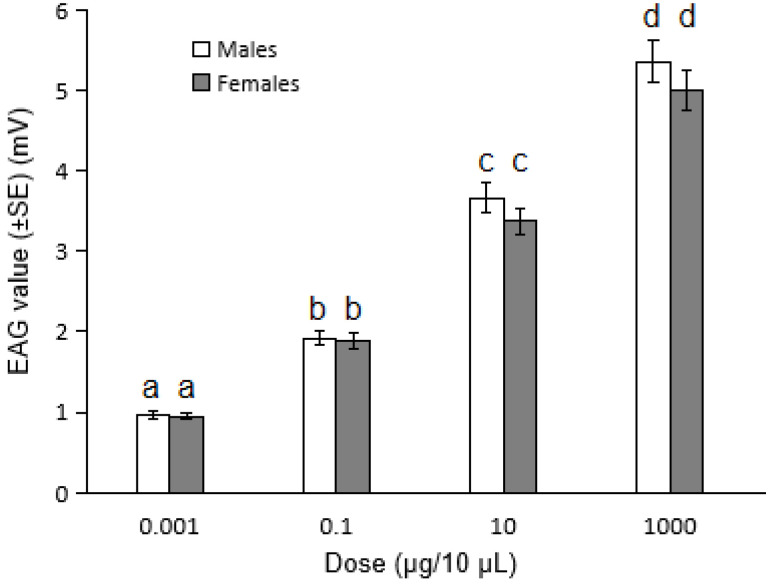
EAG responses of male and female *S. granarius* to 3-hydroxy-2-butanone at different concentrations (*n* = 9). Different letters (a, b, c and d) above bars indicate significant difference in EAG value at different concentrations (*p* < 0.05, one-way analysis of variance, Duncan’s test).

**Figure 5 insects-13-00478-f005:**
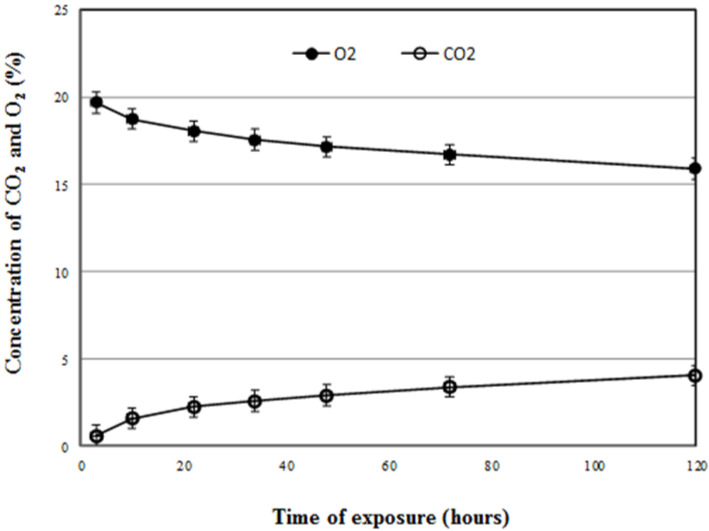
Concentrations of carbon dioxide and oxygen during five days of exposure at 25 °C (error bars indicate the standard deviation, *n* = 10).

## Data Availability

The datasets during and/or analyzed during the current study are available from the corresponding author on reasonable request.

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
