# Peer review of "The Correlation between Volatile Compounds Emitted from Sitophilus granarius (L.) and Its Electrophysiological and Behavioral Responses"

_insects, 2022, doi:10.3390/insects13050478_

Round 1

Reviewer 1 Report

The author's work titled “The correlation between volatile compounds emitted from Sitophilus granarius (L.) and its electrophysiological and behavioral responses. In this study the volatile organic compounds from Sitophilus granaries were collected and analyzed. The compounds, 3-hydroxy-2-butanone and 1-pentadecene, were identified from mixed genders of weevil.

The research goals were to collect and characterize the volatile organic compounds released from the stored grain pest under various conditions using SPME. The compounds were identified using GC-MS. EAD and bioassays were used to evaluate the behavioral response of the weevils.

Overall, the manuscript was well written, and this research can be replicated with the information provided. The author's methodology shows credibility that these compounds may be being produced by the weevils. The methods and assays developed are suitable to investigate the author's research questions. The data presented are sound and do justify the author's conclusion, supporting the claim that the granary weevils may release volatile chemicals that are acting as pheromones. The analysis and interpretation of the author's finding are simplified and comprehensible. Overall, the manuscript was well prepared, and this study can be replicated with the information provided. Therefore, this work has a potential broader application and will contribute well to the literature on stored product pest infestations

Abstract:  Well written.

Line 55: Should be “Restricted”.  Restricted where?  In what countries?

Line 80:  This sentence is not needed here.

Line 164: The open arena olfactometer was a modified version of a walking bioassay previously used for observing beetle behavior as previously described before

Line 233: S. granaries

Line 255: What solvent was used in in the assays for the different concentrations

Line 280: are was

Line 284: S. granaries

Author Response

Reviewer 1

  1. Line 55: Should be “Restricted”.  Restricted where?  In what countries?

Changed and added “in the global market”

  1. Line 80:  This sentence is not needed here.
    Deleted!
  2. Line 164: The open arena olfactometer was a modified version of a walking bioassay previouslyused for observing beetle behavior as previously described before
    Agree and deleted
  1. Line 233:  granaries
    Changed to “S. granarius
  2. Line 255: What solvent was used in in the assays for the different concentrations.
    Hexane. Added
  1. Line 280: are was
    Changed!
  2. Line 284:  granaries
    Changed!

Reviewer 2 Report

The manuscript by Cai et al. investigates the volatile compounds emitted by the granary weevil Sitophilus granarius (L.) adults at high population density and the potential behaviour-regulating functions of the two identified major components. The experimental idea is interesting and the experimental design is well constructed but, in my opinion, in its present form the paper cannot be accepted and needs of a deep revision before a re-submission.

Major concerns

The aim of the research has to be well addressed and more clearly defined, preferably at the end of introduction.

More attention has to be paid to the data analysis and results presentation:

  • in the results section regarding the volatile profile of S. granarius  there is not a match to the five replicates of volatile compounds collection as stated in the materials and methods (line 127 and again line 140), or at least to the results variability among replicates, in quantitative terms;
  • amount of chemical not only increased with increasing exposure time in the following 4 h, but also with the increasing numbers of weevils (lines 244-245): how much? even as percentages;

  • some peak intensities varied dramatically from each other at 42−44 h (lines 249-250): again, how much?

  • by the GC chromatograph it is clear that some other peaks could be identified, or they refer to unknown compounds? it would be very useful to know;
  • figure 2 is wrong; the y axis should indicate the RI and not the mean number of beetles attracted, I suppose. However, I suggest to replace the figure with a table reporting the real number of responding insects and statistical analysis; in M&M (line 222-226) is stated "The significance of the mean RI in each treatment of the two-choice pitfall bioassay was evaluated by the Student’s t-test for paired comparisons. The most significant positive or negative mean values of RI were first analysed by an analysis of variance and subsequently ranked by the least significant difference (LSD) multiple range test (p = 0.05)" but there isn't any result of these analyses;
  • again, in the EAG results there isn't any result referred to Dunnett’s (comparison with a control) or Newman and Keuls’ (comparison by groups of means) tests stated in M&M (line 229-230);
  • about figure 3, it is necessary to explain (in M&M section) how EAG responses were normalised;
  • in my experience with similar compounds, the EAG responses in mV to 3-hydroxy-2-butanone seems extremely high, and you used a custom-built amplifier: it could be, but check carefully if the necessary correction to neutralise the amplification was made;
  • it is very difficult to justify the absence of an EAG response to 1-pentadecene and at the same time a repellent effect according to behavioural results; the reference to olfactometer bioassay using another insects like Tribolium confusum (Herbst) is not enough and you need to find alternative hypothesis since to be active the compound has to be perceived.

Minor concerns

All along the text:

  • check S. granarius, sometimes not in italic (e.g. lines 233, 267, 271, 281, 284) or written as S. granaries (e.g. lines 67, 77, 83, 281)
  • change semio-chemicals with semiochemicals (e.g. lines 18, 19)
  • change electro-physiological with electrophysiological (e.g. line 222)
  • add author to scientific names (e.g. lines 79, 81)
  • delete (L.) (line 104)
  • chenge Tribolium castaneum with T. castaneum

Author Response

  1.  

Round 2

Reviewer 2 Report

The paper, particularly the presentation of results, was improved and can be accepted for publication after minor text editing as follows:

  • line 77, 152, 186, 232: change beetle/beetles with weevil/weevils;
  • line 237, 254, 281: S. granarius in italic;
  • Please note: check for consistency lines 184-185 with lines 188-189.
  • Please note: add a caption to figure 5

Author Response

Reviewer 2 

  1. line 77, 152, 186, 232: change beetle/beetles with weevil/weevils;
    Changed “beetle/beetles” to weevil/weevils.
  2. line 237, 254, 281: S. granarius in italic;
    Changed “ granarius” to “S. granarius”.
  3. Please note: check for consistency lines 184-185 with lines 188-189.
    Revised to make it consistent.
  4. Please note: add a caption to figure 5.
    Added “Figure 5. Concentrations of carbon dioxide and oxygen during five days exposure at 25 °C (Error bars indicate the SD, n=10).”.